# Representations of Research among Newly Graduated Paramedical Professionals: A Qualitative Study

**DOI:** 10.3390/ijerph182111331

**Published:** 2021-10-28

**Authors:** Evelyne Decullier, Mathilde Chauliaguet, Arnaud Siméone, Julie Haesebaert, Agnès Witko

**Affiliations:** 1Hospices Civils de Lyon, Pôle Santé Publique, Service de Recherche et Epidémiologie Cliniques, F-69003 Lyon, France; julie.haesebaert01@chu-lyon.fr; 2Institut des Sciences et Techniques de la Réadaptation (ISTR), Université Lyon 1, CEDEX 03, F-69373 Lyon, France; agnes.witko@univ-lyon1.fr; 3Institut des Sciences et Pratiques d’Education et de Formation (ISPEF), Université Lyon 2, F-69007 Lyon, France; mathilde.chau@gmail.com (M.C.); arnaud.simeone@univ-lyon2.fr (A.S.); 4INSERM Unit U1296 Radiation: Defense, Health, Environment, Léon Bérard Center, F-69008 Lyon, France; 5Research on Healthcare Performance RESHAPE, INSERM U1290, Université Lyon 1, CEDEX 08, F-69373 Lyon, France; 6Dynamique du Langage (DDL), Centre National de la Recherche Scientifique (CNRS), UMR 5596, Université Lyon 2, CEDEX 07, F-69363 Lyon, France

**Keywords:** paramedical professional, paramedical research, qualitative study

## Abstract

Despite a keen interest in clinical research, most paramedical professionals are unwilling to play an active role. Our objective was to explore paramedical professionals’ representations of research. Using an existing database of final year paramedical students (speech therapy, occupational therapy, psychomotricity, audiometry, physiotherapy, orthoptics), we deployed a qualitative approach composed of two successive steps: (1) a free word association task, and (2) semi-structured individual interviews. Out of the 54 students who agreed to be contacted, we received 21 responses to the free word association questionnaire, and 11 interviews were conducted. The hierarchical evocation matrix revealed that the scientific representation of research is based on words defining the research and the purpose of the research. “Collaboration” was identified as being an essential part of the research process. The central core of the representation is coherent with all its components perceived as positive. The content analysis of the interviews showed a polarization around two key points: (1) participants are interested in accessing and using evidence in their practice (2) but feel less confident about and/or motivated to generate evidence themselves. This study highlights the need to develop more research-friendly environments, especially in training institutions.

## 1. Introduction

The emergence of an evidence-based culture requires a critical mass of healthcare professionals in a position either to conduct research or to implement scientific findings [1]. However, it has been found that in Europe, less than 40% of articles by paramedical professionals reported clinical research [2]. It seems that there is no or insufficient motivation for those in paramedical professionals to conduct research; however, we do believe that although paramedical professionals may need help from research professionals, they could become just as involved in research as physicians. Their role in research must therefore be strengthened and encouraged, since these professionals are at the heart of the daily problems of patients and the healthcare system. Moreover, the translation of paramedical research projects’ results into practice has the potential to improve the quality of care.

In order to better understand this phenomenon, a survey of French paramedical students was conducted to investigate their perceptions of research. The paramedical professions targeted by the survey were speech therapy, occupational therapy, psychomotricity, audiometry, physiotherapy and orthoptics since they all work towards the same specific objective of patient rehabilitation, and their needs in terms of research capacity building are likely to be similar [3]. This survey revealed that 98.6% of the respondents found research to be useful, but only 34.3% were willing to conduct research themselves [4]. It also found that the most important barrier to research was the professionals’ preference for their core work. This positive attitude towards research has been reported in numerous studies of physiotherapists in European [5], American [6] and North African [7] contexts. The issue of the importance of core work has also been raised with regard to nurses [8] who share the same mission as other paramedical professionals, i.e., providing patients with clinical services and care. This importance of core work should be considered alongside the strong interest in evidence-based practices, as one study found that up to 73.6% of speech therapy students felt they were competent in implementing at least some of these practices [9]. The discrepancy between the positive attitudes and involvement in research might be explained by the fact that paramedical professionals do not see research as part of their role or that their training does not enable them to get involved in research.

In France, paramedical professionals are trained in different contexts (university hospitals, medico-social institutes, private clinics or private practices) corresponding to different types of future employment that may also condition their perceptions of research. The training usually ranges from 3 to 5 years according to profession and does not integrate research, or it does but to a minor extent. However, the process of convergence of higher education systems in European countries (Bologna agreements [10]) is gradually bringing paramedical training into a system of universitarization in which the notion of research is introduced. Once graduated, these professionals are supervised by clinical managers and doctors, and the tasks they are authorized to perform are defined by laws. More recently, cooperation protocols have been established, allowing the implementation of derogatory acts.

System-level drivers have been mobilized over the last decade with a whole host of new grants dedicated to paramedical research made available at the national [11] and local levels, and new training and status including research tasks for paramedical professionals, and national university committees have been created to promote professorial positions for paramedical professionals. The national funding scheme, called “PHRIP”, has generated considerable interest among French paramedical professionals. They have finally identified a public funding program specifically dedicated to them, which allows them to invest themselves in their own applied research [12].

In light of these observations, we decided to investigate paramedical professionals’ perceptions of their role and the effect this perception can have on their practices, by conducting a study to investigate professional representations. Professional representations are a way of understanding individuals and groups by analyzing how they represent themselves as professionals [13,14,15]. They are a particular form of social representation carried by groups linked to a trade or a professional function, and relating to objects or individuals belonging to the same sphere of professional activity [16]. Theories of social or professional representations have long been used to understand how certain health professionals understand diseases or health [17], or objects and more specific phenomena associated with their activities, such as screening or vaccination [18,19] or broader notions such as pain or the caregiver/patient relationship [20]. These representations allow group members to recognize and communicate with each other, but also to differentiate themselves from the rest of the population or other professional groups—in other words, to develop a professional identity [21]. Social representations guide people’s behavior, practices and attitudes. Professional representations have the same function as social representations, but applied to a professional context. They develop through professional action and interaction, providing a common framework for understanding the reality of the people concerned, guiding their practices and enabling them to know how to behave, enabling them to justify their past or future actions and finally contributing to establishing a professional identity [22]. There is a consensus that these professional representations are formed, in part, during the professional’s training or apprenticeship. The literature therefore speaks of socio-professional representations [23]. These representations are not only social representations because they include technical aspects specific to the targeted profession, explored during training, nor can they be considered as only professional because they do not yet incorporate enough of the experience that forms the identity and the shared memory of the professional group [16]. Distinguishing social, socio-professional and professional representations makes it possible to trace the professionalization process, as the individual’s status changes from member of the general public to student, and finally to professional [24]. Here, we hypothesized that the paramedical professionals’ representation of research would be globally positive and largely impregnated with very formal elements addressed during their training, relating to techniques or methodological approaches. However, we also supposed that the periphery of the representation would contain elements about the difficulties to implement these steps within the framework of a paramedical activity, which would reflect their new professional integration.

This qualitative methodology was deployed to explore paramedical professionals’ representations of research and to understand the barriers and facilitators to engaging in the research process. The ultimate goal was to define which factors would create the most favorable environment for those interested in conducting research.

## 2. Materials and Methods

### 2.1. Framework

Following the initial work of Moscovici in 1961 [11,12], Abric developed central core theory in 1976 [9]. Amongst the different methods for collecting social representations, the method recommended by Abric [15] is the free association method followed by a posteriori ranking of the cited elements [15,25]. Themes which are frequently cited and of high importance are considered to be the “central core” of the representation [15]. According to this theory, a representation is composed of several parts. The central core is consensual and collectively shared. It is characterized by a coherence and stability that allows it to resist change. The so-called “peripheral” elements are organized around the central core. They are more unstable and less prominent. The first periphery (low importance but frequently cited) allows individuals to anchor the representation in reality and contains more tangible items. The contrast zone contains items of high importance but infrequently cited, which can be the marker of a subgroup presence. Finally, the second periphery contains the infrequent and unimportant elements, and supplements to the first periphery.

Furthermore, in order to fully explore professional representations, Moliner recommended supplementing the word association task with other methods such as semi-structured interviews, which, through content analysis, can be used to study the participants’ discourse which includes their personal experience [26].

### 2.2. Design

We used a qualitative approach composed of two successive steps: (1) free word associations, and (2) semi-structured individual interviews. The report on the results of our research complies with the Standards for Reporting Qualitative Research (SRQR) guidelines [27].

### 2.3. Participants

In a previous study, we investigated the perceptions of research of paramedical students in their final year (speech therapy, occupational therapy, psychomotricity, audiometry, physiotherapy, orthoptics) through a national online survey conducted between January and March 2019 [4]. At the end of the survey, participating students were asked to provide their email address if they were happy to follow up with an interview in order to explore their perceptions in more depth.

All the participants that provided an email address were included. On 24 April 2020, they were sent an email inviting them to fill out a short questionnaire (free and ranked associations) and to set up an interview. By the time the qualitative study was conducted, some of the students had started work.

### 2.4. Researchers

The data collection (free word association task and interviews) was conducted by MC, a trainee in qualitative research with a background as a dietician, someone the participants could identify with (young professionals in the paramedical field). The data analysis was also carried out by MC, closely supervised by AS, who is a social psychology researcher. All authors were involved in the data interpretation.

### 2.5. Free Word Associations: Data Collection and Data Analysis

In the first step, all survey participants were asked to evoke 5 to 10 words they associate with the word “research”, and they also had to set a tone (negative, neutral, positive) for each of these words. In the second step, participants were asked to rank these words according to the importance of their link with the concept of research.

The collected material was then analyzed using the Iramuteq software (“Interface de R pour les Analyses Multidimensionnelles de Textes et de Questionnaires”, freely available at http://www.iramuteq.org/ May 2020), initially by grouping together evoked terms with the same meaning so that they can be processed. The whole set of words was then described in terms of the frequency of words, stability of the corpus (assessing the consensus on representation), variability across participants and the polarity (positive/negative connotation) and neutrality (Table 1) [28], using the Iramuteq software. The cross-tabulation of the frequency and importance rankings was then used to construct the hierarchical evocation matrix [15].

### 2.6. Interviews: Data Collection and Data Analysis

All volunteers were included in the interview phase until thematic saturation was reached; we anticipated that 12 to 15 interviews would be sufficient to reach saturation, as reported in the literature [29]. When the participants agreed to an interview, a telephone/videoconference meeting was scheduled according to their availability (the interviews could not be conducted in person due to the ongoing pandemic). The first step was to present the communication contract, explaining the interview objectives and methods (recording of the interview), and the interviewee’s rights (anonymity, freedom to participate, recording). If the participants still agreed to the interview, it could then start.

The interview grid is provided in a Appendix A and was built based on results of our previous survey [4], and the results of the free word associations. It was structured around 3 notions: the role of research and evidence in their professional practice, the role their professional context plays in implementing research and the role of research in professional identity. The last part was composed of questions on how to increase research and the use of evidence in paramedical practice. No specific definition of “evidence” was provided, so the volunteers used their own interpretation of “evidence”.

All interviews were audio recorded, and verbatim responses were transcribed. A thematic analysis [30] was conducted to identify, analyze and report patterns (themes) within the data using Nvivo Software (QSR International (1999) NVivo Qualitative Data Analysis Software, available at https://qsrinternational.com/nvivo/nvivo-products/ May 2020). For this purpose, we conducted an inductive and deductive analysis using an analysis grid which was first designed around the themes of the interview grid and then enriched and reframed based on the interview data. The quotes were labeled using acronyms indicating the participant’s profession and interview number (ST speech therapy, OT occupational therapy, PT physiotherapy, PM psychomotricity).

## 3. Results

### 3.1. Participant Profiles

Out of the 791 survey participants, a total of 54 students agreed to take part in the next stages of the study by providing their email address. After excluding 6 incorrect email addresses, 26 non-responses, 1 refusal and 1 incomplete response, there were 21 responses to the free word association questionnaire. Twelve people agreed to an interview, but one record was damaged, and therefore there were eleven interviews available.

These two samples appear to be similar in composition to the initial survey (Table 2).

### 3.2. Free Word Associations (n = 21 Participants)

A total of 151 words were evoked, representing a mean number of 7 words per participant. Overall, 98 words were distinct (65%): there were 66 hapaxes (words cited only once) and 32 words cited more than once. This yielded a stability index of 0.65, meaning that the set of words had high lexical diversity (see box 1 for definition). This diversity is essentially linked to the heterogeneity of the corpus (variability index of 0.67). These indexes reveal that the representation of research is not stable across paramedical professionals. The most commonly cited word was “collaboration” with five citations, followed by “evidence”, “rigor”, “advances”, “long”, “bibliography” and “knowledge” each with four citations.

Furthermore, the neutrality index of -0.66 and the polarity index of 0.57 indicate that the representation of research is not neutral and tends toward positivity (Table 1).

The free and ranked association matrix revealed that the scientific representation of research is based on words defining the research and the purpose of the research (central core in Table 3). “Collaboration” was identified as being an essential part of the research process. The central core is coherent with all its components perceived as positive. Overall, the professional representation is that research is based on a scientific methodology composed of an experimental component to prove a hypothesis in order to advance and develop knowledge or understanding.

The first periphery anchors this representation in reality in relation to the notion of time, publications and training. The contrast zone seems to identify a subgroup of participants who highly value research and highlight its nobility (discovery, knowledge, passion).

The second periphery reinforces the first periphery by acknowledging that research is difficult and requires rigor, but the results are useful.

### 3.3. Interviews (n = 11 Participants)

The interviews were held by phone (*n* = 3) or videoconference (*n* = 8) due to the ongoing pandemic. They lasted 58 min on average (minimum 34, maximum 80 min).

Nine themes were identified: facilitators for research (representing 31% of the corpus), barriers to research (25%), training (24%), professional identity (22%), definition of research (19%), attitudes and feelings towards research (18%), skills and knowledge (12%), objectives and consequences of research (14%) and other (5%). For each theme, the most frequently cited sub-themes are presented in Table 4.

All the themes and sub-themes seem to polarize around two key points: (1) participants are interested in accessing and using evidence in their practice (2) but feel less confident and/or motivated to generate evidence themselves. For each theme, verbatim responses from the interviews are presented in Table 5.

#### 3.3.1. Use of Evidence-Based Practices

Although less than the half of the participants stated that they use evidence when making a clinical decision, access to evidence appears to be a major issue in numerous themes (facilitators, barriers, training, skills and competence). Having access to research results and evidence is a major facilitator which encourages paramedical professionals to use evidence in their practice. They also regret that this access is limited at the end of their studies. In order to build scientific literature into their practice, they need to be able to search, find and analyze articles.

#### 3.3.2. Patient-Centered Practices

The professional identity of the paramedical professionals working in rehabilitation shows that they are first and foremost patient centered. Research has its place alongside their clinical practice but is not a priority and is not considered to be a proper part of their role. Even the research objectives are seen through the prism of the patient: research is seen as a way to provide patients with the best possible care. Some paramedical professionals report that certain physicians consider that paramedical professionals should not lead research.

#### 3.3.3. Attitude and Feelings towards Research

The spontaneous expression of the attitude and feelings toward research was not polarized (negative, neutral or positive) but remained very contrasted with positive as regards the outcomes of research and more negative regarding conducting research. Indeed, generating evidence is perceived as complicated and time consuming. It is sometimes perceived as a constraint/obligation. Some participants felt passionate, hopeful and optimistic about research, whereas others felt disappointed, uninterested and harassed with a feeling of incompetence. Whatever the positive or negative feelings, the need to be surrounded by people involved in research was evoked in relation to the themes of facilitators, barriers, training and identity.

#### 3.3.4. Training and Context Influence

Another key point concerned the research setting for paramedical professionals. According to the participants, the educational environment (attitude toward research, promotion of research, research training, references to research during other courses) is perceived as a more important facilitator than the training courses themselves. However, teachers and lecturers lack research training and experience and rarely discuss practical examples.

The participants felt that, in France, very little paramedical research is conducted and that there is a corresponding lack of paramedical researchers to act as role models to provide guidance and support. Many supervisors or colleagues do not use evidence in their practice, do not conduct research, do not want to conduct research and do not support paramedical researchers. This is even more important for professionals who have followed a revised curriculum, since they feel at odds with many professionals trained under the old curriculum. Moreover, according to some professionals, institutions/trade unions remain fixed in their way of thinking, blocking the development of practices and professions.

## 4. Discussion

### 4.1. Professional Representations of Research

Among newly graduated paramedical professionals, the central core of the professional representation of research contains several general elements (science, methodology, experimentation) relating to a very academic anchoring of this representation. Nevertheless, the periphery also presents some elements (time, difficulties, training) underlining the presence of potential obstacles to its implementation which can be the result of the confrontation with the professional reality.

Access to evidence is cited in most themes, with paramedical professionals insisting on the fact that access must be facilitated, and that knowledge is required to use evidence correctly. Overall, the newly graduated paramedical professionals perceived themselves more as users than as generators of evidence. Research is also essentially perceived as patient centered; this should be considered in light of the fact that the main barrier to research was a preference for their core work [4]. This also concurs with the fact that nurses’ satisfaction at work has been found to strongly correlate with thinking of their function in terms of “people-centered” care of patients [8], and the fact that research has been perceived as not being part of their culture [31].

The participants also raised the difficulties to engage in an academic career, due to the lack of dedicated PhD programs in their disciplines, and of dedicated academic positions. There are multiple difficulties: difficulties in finding a relevant and motivating subject for their PhD, difficulties in adapting their professional activity and, finally, the absence of positions with mixed care research activity as it is proposed for physicians. This is changing in France with the recent creation of three new national university committees in maieutics, rehabilitation sciences and nursing sciences, and the first nominations of paramedical professionals to academic positions, but it is still exceptional and seen as a path requiring sacrifices, notably towards clinical/care activity. The PHRIP funding scheme could also be a lever since it has been shown that it has generated a tendency towards interventional and quantitative studies [12].

### 4.2. Collaboration as a Facilitator

The notion of collaboration and being supported by competent professionals was a common thread throughout the interviews: collaboration is seen as a necessity, and its absence is experienced as a barrier to engaging in research. This is in line with the finding that paramedical professionals cannot envisage the idea of participating in a research process that is not supervised by a physician [31].

In contrast, the most striking difference in the central core is that clinical managers place an emphasis on autonomy [32]. The possible co-existence of different representations for the same object according to the different status in the organization is well documented [33]. Indeed, the content of professional representations is largely associated with the positions of the professionals interviewed.

In their clinical practice, nurses are constantly striving to obtain the right to work more independently of physicians, which might explain the emphasis clinical managers place on the need for autonomy [8]. The situation is no doubt very similar for all paramedical professionals. The need for collaboration in research may therefore conflict with the need for autonomy in daily work.

### 4.3. Barriers to Research

The overall environment is considered to be discouraging. Indeed, the participants insisted on the lack of experienced supervisors both during their studies and their internships.

Our qualitative approach confirms the lack of confidence and/or motivation to conduct research, but the participants also highlighted the unfavorable environment. This unfavorable environment has also been evidenced in another study in which line managers were found to consider paramedical research as utopic [32]. A lack of confidence has also been demonstrated for evidence-based practice (EBP): despite an increase in knowledge about EBP over the years, students’ EBP self-efficacy and task value have not been impacted [34].

### 4.4. Limitations

The main limitation of this study is the choice of newly graduated paramedical professionals (as professional representations may well change over time). Indeed, the diversity of words observed in the free association questionnaire might be a sign that the professional representation of research is not yet mature. Furthermore, there were no specific questions to distinguish between the perception of science versus the perception of research. However, science and research seem to be deeply intertwined since the professional representation is that research is based on a scientific methodology.

Our study was also limited by the number of participants who agreed to participate. It is therefore not possible to attest that thematic saturation was reached. Given the diverse profiles of the paramedical professionals interviewed, the only way to ensure this would be to interview several professionals each with the same profile. However, the data obtained from the interviews with the various participants converged and made it possible to identify representations that are shared across the different profiles.

### 4.5. Perspectives

Our study highlights the need to develop a more research-friendly environment, where paramedical professionals are encouraged and supported to get involved in research, especially in training institutions. This shift might be the only way to influence paramedical professionals’ representation of research. Indeed, according to Abric [35], representations can only change through the modification of practices. However, these modifications will only come about if the new practices are gratifying and do not interfere with pre-existing values, which, as it would appear, is currently the case. In addition, in order to cause a shift in representations, these practices have to be recurrent. This change in representation is the requisite first step towards the full implementation of research in the paramedical field. Some papers have addressed the issue of promoting allied health research, particularly in the United Kingdom [36]. In France, system-level drivers have been mobilized over the last decade with a whole host of new grants dedicated to paramedical research being made available at the national and local level, and new training and status including research tasks for paramedical professionals, and national university committees have been created to promote professorial positions for paramedical professionals.

## 5. Conclusions

This study explored paramedical professionals’ representations of research and the barriers and facilitators to engaging in the research process. The ultimate goal was to define which factors would create the most favorable environment for those interested in research. We found that the participants insisted on the lack of experienced supervisors both during their studies and their internships as an important barrier, and that managers might need to accept more collaboration.

## Figures and Tables

**Table 1 ijerph-18-11331-t001:** Definitions for the lexical analysis.

	Usefulness	Formulae	Interpretation
**Stability index** [25]	Assess the consensus on the subject	Nb of different words/nb total words	Close to 0: consensus on the representation Close to 1 absence of consensus
**Variability index** [25]	Assess the inter-individual variability	Nb hapax */nb of different words	Close to 0: homogeneity of participants Close to 1: heterogeneity of participants
**Polarity index** [28]	Assess the overall connotation of the set of words	(Nb positive words-Nb negative words)/nb total words	−1.00→−0.05 negative connotation −0.04→0.04 equal 0.04→1.00 positive connotation
**Neutrality index** [28]	Assess the overall neutrality of the set of words	(Nb neutral words-(Nb positive words-Nb negative words))/nb total words	−1.00→−0.05 low neutrality −0.04→0.04 medium 0.04→1.00 high neutrality

* Hapax: word with only one occurrence.

**Table 2 ijerph-18-11331-t002:** Profile of participants.

Variable	Value	Original Survey	Evocation Matrix Respondents *n* = 21	Interview Respondents *n* = 11
Sex	Male	116 (15)	2 (10)	1 (9)
	Female	675 (85)	19 (90)	10 (91)
Age	n	791	21	11
	Mean(std)	24.4 (4.66)	27.6 (5.3)	28.0 (6.0)
	Median	23	25	25
Profession	Audiometry	11 (1)	0 (0)	0 (0)
	Occupational therapy	208 (26)	5 (24)	3 (27)
	Physiotherapy	270 (34)	7 (33)	4 (36)
	Speech therapy	182 (23)	5 (24)	2 (18)
	Orthoptics	59 (7)	1 (5)	0 (0)
	Psychomotricity	61 (8)	3 (14)	2 (18)

**Table 3 ijerph-18-11331-t003:** Free and ranked association matrix.

Category * (Frequency; Importance; Polarity)		Importance Ranking (Lowest Rank = Highest Importance)	
Citation frequency		High importance (≤4.2)	Low importance (>4.2)
	High (≥7)	Central core	First periphery
		Methodology (20; 3.9; +) Advance (11; 3.6; +) Evidence (9; 3.6; +) Collaboration (10; 4.2; +) Science (9; 4.2; +) Experimentation (7; 3.4; +)	Diffusion (15; 4.9; +) Time (10; 6; n) Training (9; 6.8; n)
	Low (<7)	Contrast zone	Second periphery
		Knowledge (6; 3.7; +) Passion (6; 3.8; +) Discovery (5; 3; +) Thinking (5; 3.6; +) Interpretation (4; 3.5; +) Population (3; 2; +) Theme (2; 1.5; +)	Rigor (6; 4.3; +) Difficulty (5; 4.8; -) Egocentric (3; 5.7; -) Competition (2; 4.5; -) Useful (2; 7.5; +)

* The category includes the word itself and its different formulations. +, - and n correspond to the tone attributed to each word (positive, negative, neutral).

**Table 4 ijerph-18-11331-t004:** Themes and most frequent sub-themes extracted from the 11 interviews.

**Facilitators**		
Access to evidence	Collaboration	Guidance from people or structures specialized in research
**Barriers**		
Access to evidence	Conducting and framing a research project	Time
**Training**		
Promotion of experience and observations	Training increases research and use of research	School environment
**Professional** identity		
Patient-centered practice	Paramedical profession	Advances in practices and professions
**Definition of research**		
Methodology	Dissemination	Scientific field
**Attitudes and feelings toward research**		
Positive attitudes	Negative attitudes	Interest and motivation
**Skills and knowledge**		
Critical reading	Bibliographic search and retrieval of evidence	Communication
**Objectives and consequences of research**		
Helping the target population and improving their care	Disseminating research	Support practices, decisions, projects

**Table 5 ijerph-18-11331-t005:** Verbatim responses by theme.

Participant	Theme
*Use of evidence-based practices*	
OT1	“So I usually ask my colleagues, if I really have a doubt about the management of a patient, for their well-being, to ensure good care, I ask colleagues who work in a similar rehabilitation profession, generally I refer to other people’s opinions and if we are really up against the wall, there are Facebook groups, on social networks”
ST1	About not using evidence: “Because uh.. it takes time and in practice I don’t have any.”
OT2	“there are difficulties on several levels, i.e., access to databases, sometimes the articles are not free, so you have to be able to get them, then there is also the English language”
PT1	“knowledge is not freely available”
OT1	“you can’t pay for every article you think you might like”
PT4	“In fact, the role of research is that we have to build on it.”
*Patient-centered practices*	
PM1	“It’s not our main role either […] well, as a care giver or reeducator, the research is still apart, I think, it’s apart”
OT1	“you can sometimes get stuck with a patient, and it’s sometimes at this point, I think, that research can really help”
ST1	About what might prompt them to do research: “I think it would be a case where I really felt helpless, I would say to myself ‘we really have to do something’, so I would really look in the literature, I think. […] and if I didn’t find anything in the literature, I’ll say to myself, ‘someone has to do something’, maybe that could motivate me, I think that really if there was a case where I couldn’t find anything, it’s not normal, someone has to do something, that could be a trigger.”
*Attitude and feelings towards research*	
PM2	“there is something very energizing about research, which allows you to be or to continue to be curious and active in your work”
PM1	“If there’s a study needing a psychomotor therapist to do tests and assessments, I’ll have more fun doing that than spending hours writing a protocol* for an ethics committee, and getting turned down by the journals”
OT1	“everything is not always rosy, it doesn’t always work, but when it does, yeah it’s cool!”
PM1	“it’s still a lot of hard work; in the end, it’s nice, but it’s still quite hard work.”
PT3	About the need to get involved in research: “To have people with me, to not be alone”
ST1	“Having a team to do research is essential!”
PM1	“Because, as a paramedical professional, on my own I don’t really see how it’s possible”
ST2	“It’s not difficult to have an idea, to read an article and then that’s it, but it’s quite different to actually get started and here I think it’s essential to have a network, to have a doctor to talk to”
OT2	About the prerequisites for getting involved in research: “someone who has experience in research and who can ensure the quality of the methodology, professionals in the field, a medical reference person, a reference person for the methodology, an academic and then maybe, I don’t know, expert patients, people who are professionals in the paramedical field”
*Training and context influence*	
PM2	“I think that it is really important that teachers make use of articles, and of reliable sites, […] I really think that teaching is really important for me, seeing teachers using these data, so that afterwards we will have this reflex too”
PT1	“It’s true that even the people who teach, sometimes they weren’t at all… they were also out of their depth”
PT4	“To have people who are knowledgeable, who give us that kind of knowledge, that kind of skill”
PM1	“open up labs a bit more or have internships in research, internships even just for a few hours”
PM2	“but my internship supervisors didn’t get it, they said they were ‘not competent or not interested’”;
PT2	“my internship supervisor was not trained at all to direct a research thesis, (…) the problem today, in the world of physiotherapy, is that there are no people trained to train people and as a result, (…) I made mistakes that I would never have made if I had had a thesis director capable of guiding me on what I should do”
PT3	“but if you look at physiotherapists who graduated, let’s say, let’s broaden it to ten years, unless they are interested, ten years ago, none of them know how to do a PubMed search, most of them don’t even know about PubMed. At the hospital in the context of Covid, we talk about PubMed and they say ‘what is PubMed?’ “
OT1	citing her internship supervisor “Ah, he has done research, but I’ve been doing my job for thirty years, he’s not going to teach me how to do things”
PM2	About the difficulties in paramedical research: “Yes, and finally, there are also always these questions of hierarchy and the choice of the hospital, the management, the framework, many factors that come into play”
PM2	“there are obstacles, I would say human, colleagues who may not understand what we do and may not be supportive, it can be divisive for the team and other human obstacles, the paramedical manager who may not be favorable to research, so it can be complicated or even impossible to implement”
PM2	“there were also places where research was seen as something reserved for… doctors or people who had done theses and so “it’s not for paramedical professionals” in inverted commas “we don’t know how to do it and we don’t have the time for it, so we don’t have the skills and moreover, we don’t have the time because of the day-to-day management”
OT2	“there are colleagues who don’t want to think, who are tired of thinking about their practice, of setting up something new, I think some of my colleagues are well into their usual routine and don’t want to change anything”
ST1	“there is no institutional path for research. Well, I have a friend, he’s going to start a thesis, it’s not a thesis in speech therapy, we don’t have that, it’s an institutional, structural problem that should be remedied because it doesn’t encourage people to do research because there are no provisions in place!”
PT1	“In France, we are very behind in this respect, which is not the case in other foreign countries that have long been developing their research identity to support their profession”
OT2	“you need funding for the time you spend on research”
PT4	“it depends on the type of research (…) if I need specific equipment, I would need money”

ST1: speech therapy, woman; ST2: speech therapy, woman; OT1: occupational therapy, woman; OT2: occupational therapy, woman; PT1: physiotherapy, woman; PT2: physiotherapy, woman; PT3: physiotherapy, man; PT4: physiotherapy, woman; PM1: psychomotricity, woman; PM2: psychomotricity, woman.

## Data Availability

The individual participant survey data will not be shared.

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
