# Peer review of "Representations of Research among Newly Graduated Paramedical Professionals: A Qualitative Study"

_ijerph, 2021, doi:10.3390/ijerph182111331_

Round 1

Reviewer 1 Report

My comments are in the pdf

Reviewer 2 Report

The paper analyzes the paramedical professionals’ representations of research and identify the barriers and facilitators to engaging in the research process. It presents the continuation of the previous paper, Decullier et al. (2020), that focused on paramedical students, and now on newly-graduates, of the same participants.

While the paper, in general, is very interesting in the field of paramedical research and fits well within the aims and scope of IJERPH, it suffers from several flaws in terms of its claims and academic contribution. Therefore, the paper should be revised addressing these issues, specifically:

  1. "Paramedical research is quite a recent field and there is a general perception that it is underdeveloped." This is questionable as there are already a lot of paramedical research in the field.  The Introduction (or a separate section) should review these studies and identify the literature gap before claiming that the field is "underdeveloped".
  2. In connection to #1, there are two views on what paramedical research is: research on paramedical professionals and research by paramedical professionals. The literature shows numerous studies in the former while very limited in the latter. The reason for the latter is very obvious: there is no motivation to do research by the paramedics. Same analogy as to why farmers and laborers do a research when agriculturists and economists do the job.  
  3. " Our aims were to explore paramedical professionals’
    representations of research and to understand the barriers and facilitators to engaging in the research process." What is the study's broader aim after achieving these objectives?
  4. The study involved human subjects/participants. The paper should provide an Ethical Statement as described in the IJERPH Author Guidelines. Also, why the "Informed Consent Statement: Not applicable."? Should the participants give consent to the authors to publish their responses to the survey?
  5. The data gathering and analysis should define how "saturation" was reached. For instance, "To identify the sample size, this study followed the “data saturation” referring to a point when obtained information from participants became repetitive and the researchers would not gain any new information from further data collection <https://doi.org/10.3390/bs11050064>. 
  6. Subsection 3.5 should be renamed as "Figures, Tables and Schemes" do not really imply something relevant. Also, the results in the tables/schemes should be explained comprehensively.
  7. The Discussion section can be subdivided relative to the research objectives:  paramedical professionals’ representations of research; the barriers; and the facilitators to engaging in the research process.
  8. The Conclusion should be expanded. What was the overall aim of the paper, what are the main results? Highlight the novelty of the findings. What are their implications? What are the limitations of the study and how can these be addressed in future studies?
  9. References should be improved (increased) by including the most recent and relevant studies.
  10. Minor issues on grammar, spacing, capitalization of proper nouns, section numbering (subsections 3.2 and 3.3 missing), uniform font color in the table, Table 3 can be formatted in portrait, and Table 4 should be improved.
  11. In the Title, is the "professional" redundant? 

Round 2

Reviewer 1 Report

This second version answers the majority of comments raised by the first version, and clarifies the scope of the article. In particular, the authors added the limitation section and expanded the introduction, with a review of litterature. The clear distinction between social and professional representation is well done. They clearly explained the changes in the letter.

Nevertheless, I think there is still some issues to address to move to publication. 

1/ I would suggest to clearly identify the review of the litterature in the introduction. For the moment it appears to me strangely mixed with the context. For instance, it would be more readable to have a few more sentences on how it is important to have paramedical professionals that engage research, and then the review of the literature.

2/ Following my point 4, I still think there is a lack of context of who are those health professional, how they are trained in France (duration of the training, types of employment, etc.) and finally what kind of position they have. This seems important to me (and for the reader, especially not French) because it frames the employment profile, and consequently the relation to the research (it is different to work in an university hospital or in a private clinic). It could be added either in the introduction, in the methodology or in a small context section.

3/ Following my point 5, could you make an hypothesis based on the literature and the reality of those profession on how this attitude toward research is going to evolve when they will be employed ? You only stress that it could change…

4/ Finally, it seems that you choose not to discuss deeper the qualitative content of what “research activity” is for students. You favored to discuss the notion of evidence. But in your verbatim you have elements regarding research as a professional trajectory (with a phd) or specific position (as academic). There would be space in the discussion (but you can choose not to do it) for discussing what are the representation associated with the professional activity of research, if it is seen as a dedicated activity or embedded in the profession, and maybe make a link with the current French situation ?

Some phrasing should be modify :

  • Arab context is a weird phrasing, because before you refers to geographical area (so, North Africa ?)

  • "to implement in part" line 47

  • the notion of "purely social" feels strange even if I understand what you mean, could you change to avoid this phrasing ? (professional activity is social ...)

  • what “real life” mean line 194 (I missed it in the last version)

  • Table 3 : I don’t understand the use of the line “attitudes and feelings”, with positive and negative attitude. What does it mean to report that ?

  • “from that fact that” line 307

  • You can remove the although our objective was achieved line 312

  • line 314 it is an hypothesis, not something you demonstrated. Make it conditional as something to explore… If it is a known fact, please justify in the methodology why you are focusing on students to explore professional representations

  • line 316 : so what could be the association of research and science in your results. Please add a sentence to explicit.

  • limitations should be a dedicated section, not a sub-section of the discussion

  • I would suggest to switch 4.3 and 4.2 since 4.3 discuss more directly about research representations

Reviewer 2 Report

The authors made considerable changes to improve the manuscript. I believe it is now ready for publication.

I am looking forward to read the published version of the paper.

Author Response

We thank the reviewer for his encouragements